# CNN-Based Fault Detection for Smart Manufacturing †

**Dhiraj Neupane** [1,‡] , **Yunsu Kim** [2,‡], **Jongwon Seok** [2,*] and **Jungpyo Hong** [2,*]

1   Research and Development Department, IPCamp, Jinju-si 52818, Korea; dhirajneupane1717@gmail.com
2   Department of Information and Communication Engineering, Changwon National University, Changwon-si 51140, Korea; asd8780@gmail.com
*   Correspondence: jwseok@changwon.ac.kr (J.S.); hansin@changwon.ac.kr (J.H.)
†   This article is the extended version of the Conference Paper "Deep Learning-Based Bearing Fault Detection Using 2-D Illustration of Time Sequence", in Proceedings of the 2020 International Conference on Information and Communication Technology Convergence (ICTC). This version includes a new idea with the extension of the study presented in the conference paper.
‡   Co-first authors (Authors contributing equally).

**Abstract:** A smart factory is a highly digitized and networked production facility based on smart manufacturing. A smart manufacturing plant is the result of intelligent systems deployed in the factory. Smart factories have higher production volumes and are prone to machine failures when operating in almost all applications on a daily basis. With the growing concept of smart manufacturing required for Industry 4.0, intelligent methods for detecting and classifying bearing faults have become a subject of scientific research and interest. In this paper, a deep learning-based 1-D convolutional neural network is proposed using the time-sequence bearing data from the Case Western Reserve University (CWRU) bearing database. Four different sets of data are used. The proposed method achieves state-of-the-art accuracy even with a small amount of training data. For the sensitivity analysis of the proposed method, metrics such as precision, recall, and f-measure are determined. Next, we compare the proposed method with a 2-D CNN that uses two-dimensional image illustrations of raw data as input. This method shows the effectiveness of using 1-D CNNs over 2-D CNNs for time-sequence data. The proposed method is computationally inexpensive and outperforms the most complex and computationally intensive algorithms used for bearing fault detection and diagnosis.

**Keywords:** bearing fault; smart manufacturing; CWRU dataset; deep learning; convolutional neural network; raw vibration data

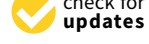

## 1. Introduction

Electrical machines are used ubiquitously in industrial applications nowadays. With the development and advancement in science and technology, modern industries are also developing rapidly. As a result, machinery equipment functions on a day-to-day basis and for almost every application, which means that these types of machinery work under unfavorable circumstances, excessive loads, and humidity. This results in motor failures, leading to massive maintenance expenditures, declines in production levels, severe financial losses, and a possible risk of loss of lives. The rotating machinery and induction engines play a vital role in the manufacturing systems. These rotating machines consist of numerous elements, such as a stator, rotor, shaft, and bearings. Rolling element bearings (REBs) are generally termed bearings and are the most vital and vulnerable components in the machine, whose fitness state affects the effectiveness and performance, stability, and lifespan of the machinery [1,2]. The four components of REBs are the ball (B), cage, inner-race (IR), and outer-race (OR). The experimental test rig of the Case Western Reserve University (CWRU) ball bearing system and the bearing components are shown in Figure 1. The bearing fault, one of the most common faults in machinery, accounts for 30% of the total faults, causing the machine to break down and eventually resulting in a severe loss of

safety, property, and even the loss of lives in some cases. Hence, bearing fault detection and diagnosis have attracted researchers and scientists and have become essential for scientific advancement [3,4]. With the growing concept of Industry 4.0 and smart manufacturing, intelligent methods for detecting and classifying machinery faults have been a key part of scientific research and interest. The fault detection system and the bearing health-state monitoring system are anticipated to provide information regarding the actual working state of the machinery equipment continuously without hampering the production line. Again, the mechanical vibration signals are considered rich sources of information for the appropriate analysis of processes related to bearing faults. These vibration signals can provide enough information about the location and type of the fault, which is helpful for fault diagnosis [1,5].

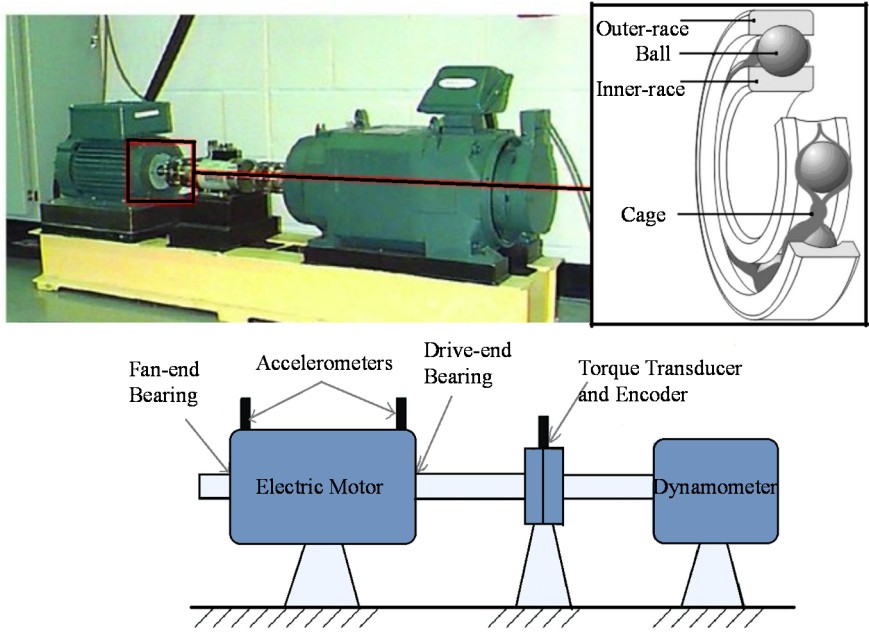

**Figure 1.** CWRU bearing test rig system and components of REB.

The working process of the ball bearing system consists of sensors placed in different locations in the equipment, via which the signals are transferred to the data acquisition system for additional processing. Figure 2 shows the vibration data collection process of the CWRU ball bearing system. The performance of fault detection methods depends both on the quality of the vibration signals collected and on the efficiency of the applied signal processing and feature extraction methods [5,6]. Traditionally, the maintenance of these REBs used to be a posterior task, usually taking place after the occurrence of the fault. Moreover, this kind of posterior maintenance procedure leads the machine to break down, resulting in financial loss and other casualties [3]. Hence, it is of great significance to surveil the bearing condition during the working state of the engine. Many signal processing, machine learning (ML), and deep learning (DL)-based methods have been suggested and implemented in bearing condition monitoring and bearing fault detection and diagnosis.

Data-driven methods use signal processing techniques in the time domain, frequency domain, and time-frequency domain to analyze vibration signals. With the use of these signal processing approaches, the appropriate height of fault detection and diagnosis accuracies were stated [7,8]. Nevertheless, these conventional signal processing methods carry some limitations. The time-domain method uses the natural properties of the vibration signals in the time domain, such as root mean square, crest factor, quadratic mean, and skewness. These characteristics can be used in dynamic system monitoring applications to effectively reflect transient machine conditions assuming a stationary signal. In actual industrial settings, condition monitoring signals are often complicated by time-varying environmental conditions such as temperature and lubricants. In addition to background

noise and interference, spectral changes and nonlinear behavior are also complicated. In other words, the actual vibration signal captured is not stationary, which limits the validity of time-domain statistics. Again, due to the weak amplitude and short duration of structural changes in the vibration signal in the initial stage, the frequency-domain approaches may be unreliable for evaluating non-stationary machine conditions. Another limitation of these methods is that they are inadequate to deal with non-stationary signals [9].

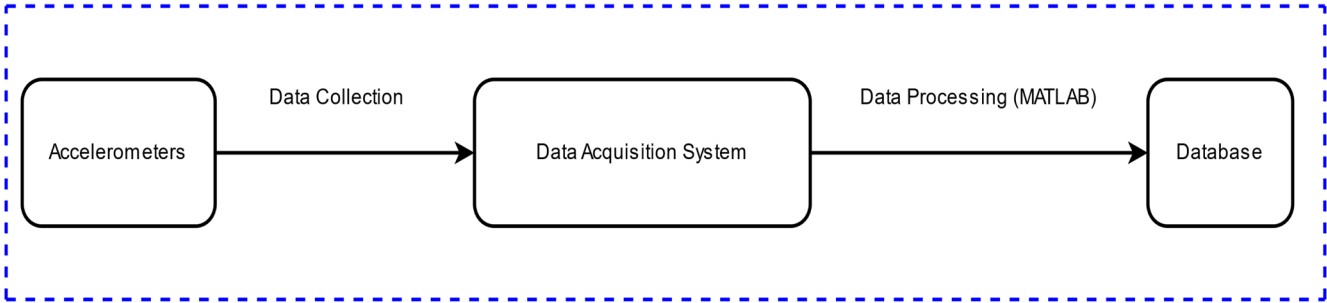

**Figure 2.** Vibration data collection process for CWRU ball bearing system.

To be more specific, temporal analysis is not capable of finding the faulty component of the machine. The frequency peak of the bearing fault is not easily distinguishable through FFT analysis. Correspondingly, cepstrum analysis is computationally expensive, and it generates many undesired large peaks near the zero point, making the output complex to interpret. The prerequisite of some experience and knowledge regarding the resonance frequency and filtering band makes envelope analysis challenging to use. Moreover, wavelet transform remains weak in the selection of an appropriate mother wavelet, decomposition level, and respective frequency band, which is necessary information for fault analysis and detection [6,10]. Again, the maxima of these methods are troublesome as they require features such as the mean, median, minimum, maximum, peak-to-peak, kurtosis, skewness, standard deviation, absolute mean, and root-mean-square (RMS) value for describing the actual bearing condition [11]. One finds it difficult to choose the exact features for analyzing the particular signal used in the classification [12]. Therefore, many ML/DL-based algorithms have been used ubiquitously for the ease of selecting the exceptional patterns present in the data, which are challenging for a human being to identify.

Machine learning, which is a subfield of artificial intelligence, can generate insights in data, even if they are not specifically instructed regarding what to search for in the data [13]. Many ML-based methods have been proposed and implemented to develop a knowledge-based architecture for the prior diagnosis of bearing faults to prevent catastrophic failure and reduce operating costs. ML-based algorithms such as artificial neural networks, principal component analysis, support vector machines, k-Nearest Neighbors, and singular value decomposition are broadly used in bearing fault detection and diagnosis. A comprehensive review of such approaches can be found in [14]. However, the problem-solving approach of ML is not satisfactory. ML algorithms first divide the problem statement into different parts and then combine the result. Alternatively, deep learning algorithms are widely accepted and implemented. Deep learning is a subfield of machine learning that defines both higher- and lower-level categories with greater accuracy. Deep learning techniques provide better efficiency and accuracy [15]. The efficient working ability of these algorithms with a huge amount of data, their end-to-end problem-solving approach, and the pleasing result has attracted many scholars [16]. DL-based algorithms are ubiquitously used in almost every field.

*1.1. Methodology*

The methodology used in this study is shown in Figure 3. First, raw bearing vibration data were collected from the CWRU bearing database, originally stored as MATLAB files in a one-dimensional format. They were then pre-processed for 1-D and 2-D CNN inputs;

for the 1-D CNN input, each sample was created from 1600 data points; for the 2-D CNN input, a two-dimensional image representation of 40 × 40 pixels was created. We then used each model to perform feature extraction and classification. Finally, we performed a performance analysis and drew conclusions.

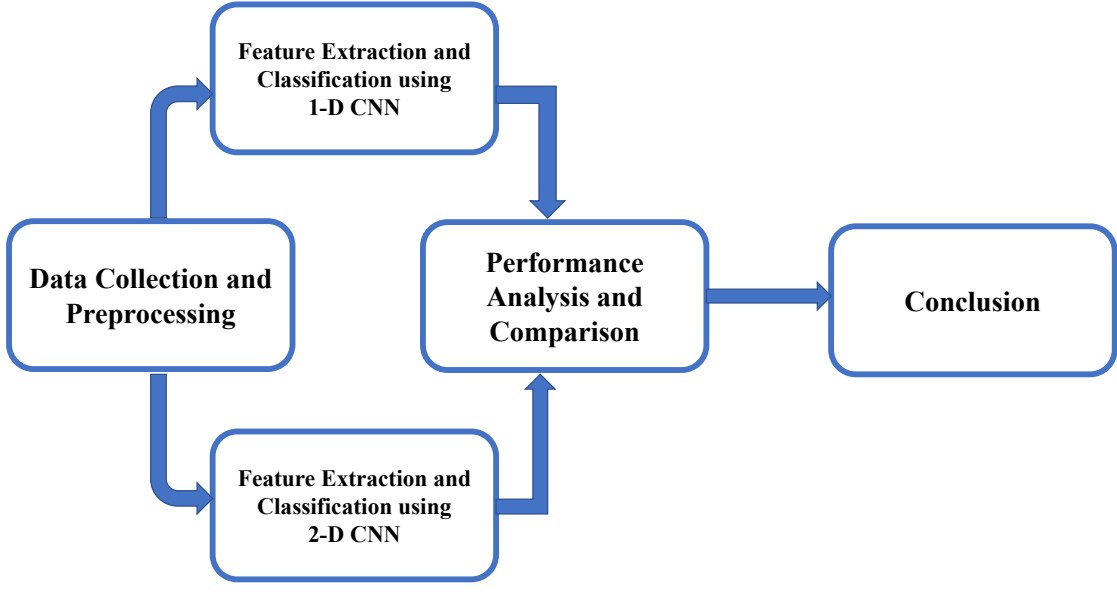

**Figure 3.** Methodology implemented.

### 1.2. Contribution and Organization

Inspired by the widespread use of convolutional neural networks, a typical deep learning model in computer vision, we used a 1-D CNN-based model to detect and classify bearing faults on CWRU time-series data. The proposed model appropriately utilizes the feature extraction and classification properties of CNNs. Thus, it is simple to apply for time-sequence data and efficient in terms of computational complexity. Furthermore, we also compared the proposed model's performance with a 2-D CNN using a two-dimensional image illustration of raw data as an input. Four different datasets were used in this research. With the use of a smaller amount of training data also, we achieved promising results. We also performed a sensitivity analysis of the proposed 1-D CNN model. Precision, recall, and f-measure were calculated, along with the accuracy, which are helpful in demonstrating the efficiency of the proposed architecture. Simplicity and computational feasibility are the main advantages of this model.

The remainder of the paper is organized as follows. A fundamental introduction to bearings and faults in bearings, conventional signal processing, and ML/DL approaches is presented in Section 1. Section 2 contains the related theory and related work—a brief explanation of the published works employing CNN and other DL-based architectures. Section 3 describes the experimental analysis using the proposed method. This section is the core section of this research. The analysis of the results and the sensitivity analysis are presented in this section. Section 4 presents the comparison between the proposed method and two-dimensional CNN using 2-D illustrated images as input. We also compares the proposed method with some of the published works as well in this section. The paper is summarized in Section 5, with a discussion and conclusions.

### 2. Related Theory and Work

#### 2.1. Related Theory

A.    Convolutional Neural Networks (CNNs)

Convolutional neural networks are extensively accepted, DL-based feed-forward networks inspired by the work of Huber and Wiesel in neuroscience [17]. LeCunn first

proposed them for image processing [18]. The CNN architectures were developed and employed widely after the significant overview of AlexNet [19], and they have replaced the outdated classification methods. A typical CNN mainly consists of an input layer, some convolutional layers, activation units, a pooling layer, and a fully connected layer [20]. The convolutional and pooling layers can be used to capture deep feature maps of two-dimensional inputs. The convolution and pooling operations and parameter sharing phenomenon of CNNs enable them to learn features from images and to run on any device [21]. The convolution operation can be interpreted as sliding a filter over data and, for each position, applying a dot product between the filter and the data at this position [22]. The convolution layers are designated for convolving the local input regions with the kernel filters, which generate the feature maps or activation maps by the activation unit. The next is the pooling or sub-sampling layer for downsampling the features and merging semantically similar features into one. This layer reduces the dimension and parameter of the network [23]. The two commonly applied pooling operations are average pooling, which determines the average value of each patch on the activation map, and maximum pooling (or max pooling), which determines the maximum value for each patch of the feature map. The next step is converting the output of the convolutional block into a one-dimensional array for inputting it into the next layer, called a fully connected layer. The last layer is the classification layer, which categorizes the objects into the respective class [24]. Figure 4a shows the general 2-D CNN model consisting of an input image and feature extraction block (convolution, activation unit, and pooling layer) followed by the fully connected layer/s and, finally, the classification stage. CNNs have immensely improved the performance and efficiency in computer vision, object detection, natural language processing, and speech recognition, with a gradual increment in the production and memory of GPU. Hence, the use of CNNs has proliferated within computer vision [13,25].

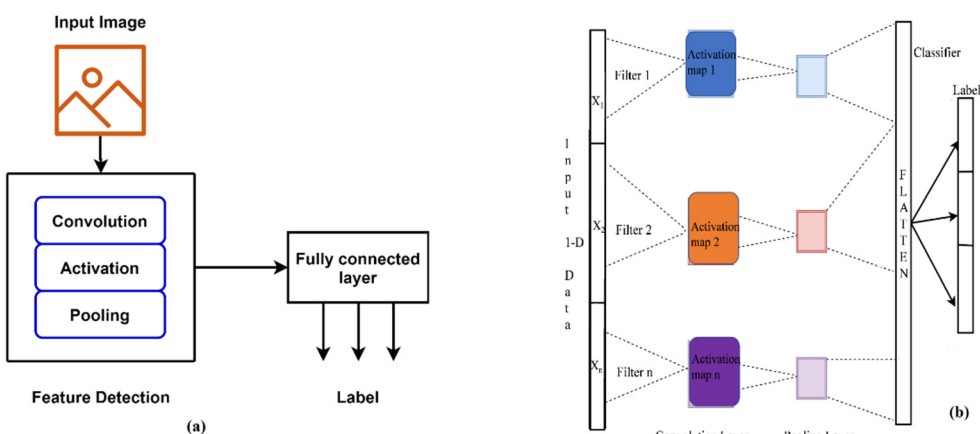

**Figure 4.** Common architecture of (**a**) 2-D CNN and (**b**) 1-D CNN.

Moreover, to deal with one-dimensional data, 1-D CNNs have been developed. In brief, 1-D CNNs are the modified version of 2-D CNNs, in which forward-propagation (FP) and back-propagation (BP) are simple array operations rather than matrix operations, which makes them more efficient for specific applications in dealing with 1-D signals [13]. In a 1-D CNN, i.e., Conv1D, the kernel slides along one dimension. In addition, the relatively shallow structure of a 1-D CNN makes it able to learn challenging tasks involving 1-D signals, and they are also suited for real-time applications [26]. Figure 4b shows the common structure of a 1-D CNN. Recent studies show that compact 1-D CNNs have provided superior performance in applications related to one-dimensional data [27]. The applications of a 1-D CNN include the analysis of 1-D sensor and accelerometer data, natural language processing, and speech recognition.

B.    SoftMax Classifier

A SoftMax regression, which is normally applied in the final layer, is a type of logistic regression used for normalizing an input value into a vector of values that follow a probability distribution whose total sums up to one [28]. In this study, we employed SoftMax regression as the bearing health condition classifier in the proposed network. It is easy to implement and quick to compute. Let us suppose that $x^{(i)}$ is the training set with their corresponding labels $y^{(i)}$, where $i = 1, 2, 3, \ldots, M$; $M$ is the total number of training samples. $x^{(i)} \in R^{M \times 1}$ and $y^{(i)} \in \{1, 2, 3, \ldots, K\}$, where $K$ is the number of labeled classes. For an input $x^{(i)}$, the SoftMax regression can predict the probability $P(y)^{(i)} = j \mid x^{(i)}$ for each label $j$, where $j = 1, 2, 3, \ldots, K$. The estimated probabilities of the input data $x^{(i)}$ belonging to each label can be obtained according to the hypothesis function,

$$
f_{\ddot{\theta}}^{x^{(i)}}
\begin{bmatrix}
P(y^{(i)} = 1 \mid x^{(i)}; \ddot{\theta}) \\
P(y^{(i)} = 2 \mid x^{(i)}; \ddot{\theta}) \\
\vdots \\
P(y^{(i)} = K \mid x^{(i)}; \ddot{\theta})
\end{bmatrix}
= \frac{1}{\sum_{k=1}^{K} e^{\ddot{\theta}_k{}^T x^{(i)}}}
\begin{bmatrix}
e^{\ddot{\theta}_1{}^T x^{(i)}} \\
e^{\ddot{\theta}_2{}^T x^{(i)}} \\
\vdots \\
e^{\ddot{\theta}_K{}^T x^{(i)}}
\end{bmatrix},
\tag{1}
$$

where $\ddot{\theta} = [\ddot{\theta}_1, \ddot{\theta}_2, \ldots, \ddot{\theta}_k]^T$ are the parameters of the SoftMax regression model. The probability distribution $P(y)^{(i)}$ can be expressed as

$$
P(y = j \mid x^{(i)} = \frac{e^{\ddot{\theta}^{(i)}}}{\sum_{j=1}^{K} e^{\ddot{\theta}_K{}^{(i)}}}
\tag{2}
$$

This classifier function ensures that the outputs are positive values ranging from 0 to 1, which are the probabilities for each class [5,29].

C. Batch Normalization

Batch normalization (BN) is a powerful tool to overcome the problem of internal covariate misalignment by adjusting the distribution of data samples before they are transformed by the activation function. This technology can significantly improve the training efficiency of deep network models [30]. Batch normalization normalizes each element of the layers of the neural network to a zero mean and unit variance based on the statistics within a mini-batch. This allows us to change the representativeness of the network so that each activation is given a learned scale and shift parameter [31]. The mini-batch-based stochastic gradient descent method is altered by computing the mean $\mu_j$ and variance $\sigma_j^2$ over the batch of each hidden unit $h_j$ in each layer, normalizing the units, scaling them with the learned scaling parameter $\gamma_j$, and shifting them with the learned shift parameter $\beta_j$:

$$
\hbar_j \leftarrow \gamma_j \frac{h_j - \mu_j}{\sqrt{\sigma_j^2 + \varepsilon}}
\tag{3}
$$

where $\varepsilon$ is a small positive constant.

D. ReLU

An activation function acts on a neuron in a neural network and is responsible for mapping the neuron's input to its output. Activation functions commonly used in ML/DL algorithms are the sigmoid, tanh, rectified linear unit (ReLU), and their derivatives. Sigmoid and tanh are saturated activation functions, which have the problem of gradient loss. ReLU is a non-saturated activation function, which solves the problem of gradient vanishing to some extent and speeds up convergence; ReLU outputs positive numbers as they are and

sets negative numbers to zero as they are. If the input is negative, ReLU will not work at all [5]. This function is defined as follows:

$$ReLU(x) = \begin{cases} x, & if\ x > 0 \\ 0, & if\ x \leq 0 \end{cases} \tag{4}$$

*2.2. Related Work*

Bearing fault detection and diagnosis through intelligent methods have become an important part of production engineering. Different ML- and DL-based supervised and unsupervised algorithms have been applied for fault detection and diagnosis. Deep neural structures such as auto-encoders [32], deep belief networks [33], generative adversarial networks [34], recurrent neural networks [35], reinforcement neural networks [36], and CNNs have generated highly satisfying results in this field. Moreover, the fusion method of ML- and DL-based and signal processing approaches is also practiced. Different graph modeling-based approaches are also frequently proposed and implemented. Some worthy mentions are [7–9,20,37–41].

Regarding the CNN architectures, both 1-D CNNs and 2-D CNNs have been employed to detect and classify bearing faults. Levent Eren proposed a 1-D CNN model in [42], which provided 97.1% accuracy. An intelligent rotating machinery fault diagnosis system based on DL using a data augmentation technique is proposed in [43]. The authors used two data augmentation methods and five data augmentation techniques, and the best testing accuracy obtained on the CWRU dataset was 99.91%. Similarly, a CNN-based approach with multiple sensor fusion is proposed in [44]. The average accuracy with two sensors was 99.41%, whereas that with only one sensor was 98.35%. In ref. [45], 2-D representation of 1-D signals is used to analyze bearing faults employing a 2-D CNN and the authors also compared their model with a 1-D CNN, where they showed that the 2-D CNN was more efficient than the 1-D CNN. In ref. [46], vibration signals, which are split into segments with the same length, are used directly as input data for the deep structure of the CNN. The amplitude of each sample in the vibration signal is normalized into the range $(-1, 1)$, which becomes the intensity of the corresponding pixel in the corresponding image. The accuracy, with 10 and 20 filters in the first and second layers, was found to be 96.75%. In ref. [47], a deep distance metric learning method is presented in which a deep CNN is used as the leading architecture. A representation clustering algorithm is proposed to decrease the distance of intra-class variations and maximize the length of inter-class differences simultaneously. A domain adaptation method is adopted to reduce the maximum mean discrepancy between training and testing data, and 99.34% accuracy was stated when the sample length was 8192. In ref. [5], D. Neupane et al. proposed a model that detects bearing failures using the continuous wavelet transform and classifies them using a switchable normalization-based convolutional neural network (SN-CNN). The stated testing accuracy was between 99.44% and 100% for different batch sizes and datasets.

## 3. Experimental Analysis

### 3.1. Data Analysis and Pre-Processing

The dataset used for this research was the CWRU bearing dataset [48], one of the most popular bearing datasets, which is publicly provided by Case Western Reserve University on their website [2]. The CWRU bearing dataset is mostly used for fault analysis and classification and detection of faulty machinery bearings; thus, it serves as a fundamental dataset to authenticate the performance of the different ML- and DL-based algorithms. The bearing test rig arrangement used in obtaining the CWRU bearing data is shown in Figure 1. It consists of a 2 hp Reliance electric induction motor, a dynamometer, a torque transducer, and control electronics (not shown in the figure). Acceleration data are collected from many sensors placed at different locations. The data are collected for normal bearings, single-point drive-end (DE), and fan-end (FE) defects. This dataset consists of 161 records grouped into four classes: 48 k normal-baseline, 48 k drive-end fault, 12 k drive-end fault,

and 12 k fan-end fault [10]. Electro-discharge machining was used to inject the single point faults into the test bearings, with fault diameters of 7 mils, 14 mils, 21 mils, 28 mils, and 40 mils. One mil is equal to 0.001 inches. Vibration data were recorded for motor loads of 0 to 3 horsepower, with motor speeds of 1720 to 1797 rpm after the faulty bearings were reinstalled into the test motor [48]. Regarding the names of the data files, the first letter represents the fault position, the next three numbers signify the fault diameters, and the last number denotes the bearing loads. For example, the data file 'B007_0' contains the ball bearing fault data, which has a fault of diameter 0.007 inches, operated under a motor load of 0 hp. Similarly, the data file 'OR014@6_1' contains the fault data of an outer-race fault of diameter 0.014 inches when the load was centered (fault in 6 o'clock position) and operated under a motor load of 1 hp [1,5].

In this research, we used the 48 k DE fault data and 48 k normal-baseline data for the experiment. The fault types used were ball fault, inner-race fault, and outer-race fault for a motor load of 1 hp, 2 hp, and 3 hp for each type of fault. Each fault type was further categorized into the respective fault of 7 mils, 14 mils, and 21 mils. We also used a normal-baseline (healthy) bearing of 1 hp, 2 hp, and 3 hp load. The total number of datasets used was four and each set was divided into 10 classes. Datasets were named dataset A, dataset B, dataset C, and dataset D. Dataset A contained the 48 k DE data of load 1 hp, dataset B contained 48 k DE data of load 2, dataset C contained that of load 3 hp, and dataset D was the combination of all datasets A, B, and C. Table 1 shows the necessary details regarding the dataset used in this research.

**Table 1.** Other necessary details of the dataset used.

| Dataset | Motor Speed (rpm) | Load (hp) | Fault Condition | | | | | | | | | |
|---------|-------------------|-----------|-----------------|---|---|---|---|---|---|---|---|---|
| | | | Ball Fault | | | IR Fault | | | OR Fault | | | Normal |
| A | 1772 | 1 | B007 | B014 | B021 | IR007 | IR014 | IR021 | OR007@6 | OR014@6 | OR021@6 | None |
| B | 1750 | 2 | B007 | B014 | B021 | IR007 | IR014 | IR021 | OR007@6 | OR014@6 | OR021@6 | None |
| C | 1730 | 3 | B007 | B014 | B021 | IR007 | IR014 | IR021 | OR007@6 | OR014@6 | OR021@6 | None |

For the data pre-processing for the 1-D CNN, each signal from the 10 classes was further divided into $N$ samples, each of 1600 points. Hence, the input for the 1-D CNN was $1600 \times 1$. Figure 5 shows the sample division process for the 1-D CNN. Thus, the total number of samples used in model training was $N \times 10$, which was further split at a train–test ratio of 0.7:0.3. Moreover, 10% of the train set was further used for the validation set. The value of N for dataset A, B, and C was 238, 303, and 303, respectively. Since dataset D was the combination of all datasets A, B, and C, it contained all the data from the earlier 3 datasets. Table 2 shows the length of the train, test, and validation sets for each dataset used in the research.

### 3.2. Feature Extraction and Classification Using 1-D CNN

For the feature extraction and classification, a 6-layer 1-D CNN was used. The network architecture contained three 1-D convolutional layers, with each layer followed by a max-pooling layer, 2 dense layers, and one final classification layer. The network architecture was 32-64-32-800-256-10. All the layers consisted of kernel filters of the size $9 \times 1$, a stride of 1, and max-pooling layers of size $4 \times 1$. The network architecture is shown in Figure 6 and the information regarding the number of parameters in each layer is shown in Table 3. The total number of parameters used in the network was 276,552. Regarding other parameters used in the network, a rectified linear unit (ReLU) was used as an activation unit for all the convolution layers, and for the classification, SoftMax was used. The GlorotNormal initializer, which is also called the Xavier normal initializer, was used for initializing the variables. This initializer draws samples from a truncated normal distribution centered on zero with standard deviation.

**Table 2.** Length of train, test, and validation set used in the research.

| Dataset | Train Set | Test Set | Validation Set |
|---|---|---|---|
| 48k_DE_Load1 (A) | 1499 | 714 | 167 |
| 48k_DE_Load2 (B) | 1908 | 909 | 213 |
| 48k_DE_Load3 (C) | 1908 | 909 | 213 |
| D (A/B/C) | 5317 | 2532 | 591 |

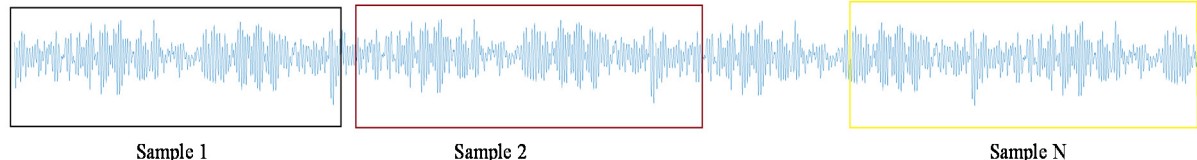

**Figure 5.** Data pre-processing for 1-D CNN.

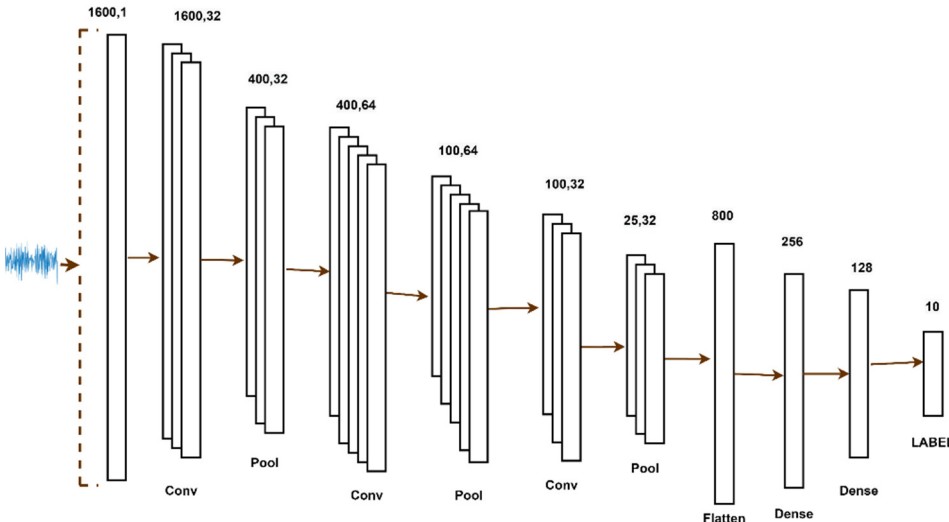

**Figure 6.** Architecture of proposed 1-D CNN model.

**Table 3.** Network architecture and parameters.

| Layers | 1-D CNN | | 2-D CNN | |
|---|---|---|---|---|
| | **Output Shape** | **Parameters** | **Output Shape** | **Parameters** |
| Input | (None, 1600,1) | 0 | (None, 40, 40, 1) | 0 |
| Conv2D | (None, 1600, 32) | 320 | (None, 40, 40, 32) | 320 |
| MaxPool2D | (None, 400, 32) | 0 | (None, 20, 20, 32) | 0 |
| Conv2D | (None, 400, 64) | 18,496 | (None, 20, 20, 64) | 18,496 |
| MaxPool2D | (None, 100, 64) | 0 | (None, 10, 10, 64) | 0 |
| Conv2D | (None, 100, 32) | 18,464 | (None, 10, 10, 32) | 18,464 |
| MaxPool2D | (None, 25, 32) | 0 | (None, 5, 5, 32) | 0 |
| Flatten | (None, 800) | 0 | (None, 800) | 0 |
| Dense | (None, 256) | 205,056 | (None, 256) | 205,056 |
| Dense | (None, 128) | 32,896 | (None, 128) | 32,896 |
| Classification | (None, 10) | 1290 | (None, 10) | 1290 |
| | **Total Parameters: 276,522** | | **Total Parameters: 276,522** | |

*3.3. Sensitivity Analysis and Model Stability*

For the further analysis of the proposed method, we applied three evaluation indices for each type of fault detection, namely precision, recall, and f1-score. Precision is the ratio of correctly predicted positive observations to the total predicted observations, recall is the ratio of correctly predicted positive observations to all the observations in the actual class, and the f-measure is the weighted average of precision and recall [49]. The precision, recall, and f-measure for each bearing class *c* are defined as follows:

$$
\begin{aligned}
Precision\ (c) &= \frac{True\ Positive}{True\ Positive + False\ Positive} \\
Recall\ (c) &= \frac{True\ Positive}{True\ Positive + False\ Negative} \\
f\!-\!measure &= 2\frac{Precision.Recall}{Precision + Recall}
\end{aligned}
\tag{5}
$$

Table 4 shows the sensitivity analysis results of the proposed model. It shows the values for precision, recall, and f1-score obtained when we evaluated the performance of the proposed model using those measures. The training, testing, and validation accuracy are highlighted in Table 5.

**Table 4.** Sensitivity analysis of the proposed model.

| Dataset | Precision | Recall | f1-Score |
|:---:|:---:|:---:|:---:|
| A | 0.9932 | 0.9932 | 0.9932 |
| B | 0.9920 | 0.9920 | 0.9920 |
| C | 0.9946 | 0.9946 | 0.9946 |
| D | 0.9949 | 0.9949 | 0.9949 |

**Table 5.** Model accuracy and loss.

| Model | Dataset | Training Accuracy | Testing Accuracy | Validation Accuracy | Average Time Taken/Sample | | Loss | | |
|:---:|:---:|:---:|:---:|:---:|:---:|:---:|:---:|:---:|:---:|
| | | | | | Train | Test | Train | Test | Validation |
| 1-D CNN | A | 100% | 99.38% | 99.33% | 119 µs | 72 µs | 2.0740e-07 | 0.0201 | 0.0068 |
| 2-D CNN | A | 100% | 96.27% | 96.0% | 168 µs | 79 µs | 8.6794e-07 | 0.2451 | 0.1609 |
| 1-D CNN | B | 100% | 99.34% | 99.53% | 111 µs | 66 µs | 9.9591e-08 | 0.0884 | 0.0117 |
| 2-D CNN | B | 100% | 97.14% | 96.24% | 140 µs | 63 µs | 3.1614e-07 | 0.2553 | 0.2013 |
| 1-D CNN | C | 100% | 99.45% | 99.53% | 111 µs | 63 µs | 2.2180e-08 | 0.0644 | 0.0130 |
| 2-D CNN | C | 100% | 97.92% | 97.18% | 115 µs | 62 µs | 2.9165e-07 | 0.2042 | 0.2614 |
| 1-D CNN | D | 100% | 99.49% | 99.83% | 110 µs | 58µs | 3.3937e-06 | 0.0130 | 0.0132 |
| 2-D CNN | D | 100% | 98.14% | 97.80% | 107 µs | 59 µs | 1.1997e-05 | 0.0997 | 0.0845 |

In order to check the model stability, we performed each experiment three times. The proposed model was found to be stable. The tolerance was found to be $\pm 0.02$.

## 4. Compared Method

*4.1. Image (2-D Representation) Construction from 1-D Vibration Data*

The raw vibration data were in one-dimensional form. The transformation of 1-D bearing raw vibration data into 2-D images is a straightforward task. The amplitude of each sample is the intensity of the corresponding pixel in the corresponding image.

The transformation between the amplitude of a sample and the corresponding pixel is represented by the following equation [46]:

$$P[i, j] = A[(i - 1) * m + j] \tag{6}$$

where $i = 1 : n; j = 1 : m; P[i, j]$ is the intensity of the corresponding pixel $(i, j)$ in the $m \times n$ vibration image. $A[.]$ is the amplitude of the sample in the vibration signal. The number of pixels in the vibration image equals the number of data points in each vibration sample. Since we took 1600 data points in each sample of vibration signal, we constructed the $40 \times 40$ equivalent 2-D image representation. In our case, $m = n$.

Here, the 1-D vibration data were divided into $N$ equal samples, with 1600 data points in each sample. This means that each sample will be an image of $40 \times 40$ pixels. Therefore, there will be $N$ images of dimension $40 \times 40$. Suppose that $S_1 = [x_0, x_1, x_2, x_3, \ldots, x_m, \ldots, x_n]$ is one sample of the 1-D signal (say $S$) among $N$ samples.

The 2-D representation of the first image matrix $y_1$ is:

$$y_1 = \begin{bmatrix} y_{11} & y_{12} \cdots & y_{1m} \\ y_{21} & y_{22} \cdots & y_{2m} \\ \cdots & \cdots & \cdots \\ y_{m1} & y_{m2} & y_{mm} \end{bmatrix} = \begin{bmatrix} x_0 & x_1 \ldots & x_m \\ \cdots & \cdots & \cdots \\ \cdots & \cdots & x_n \end{bmatrix} = \begin{bmatrix} x_0 & x_1 \ldots & x_{40} \\ \cdots & \cdots & \cdots \\ \cdots & \cdots & x_{40} \end{bmatrix} \tag{7}$$

Figure 7 shows each 2-D represented sample from the dataset 'A' for ten different health condition bearings [3].

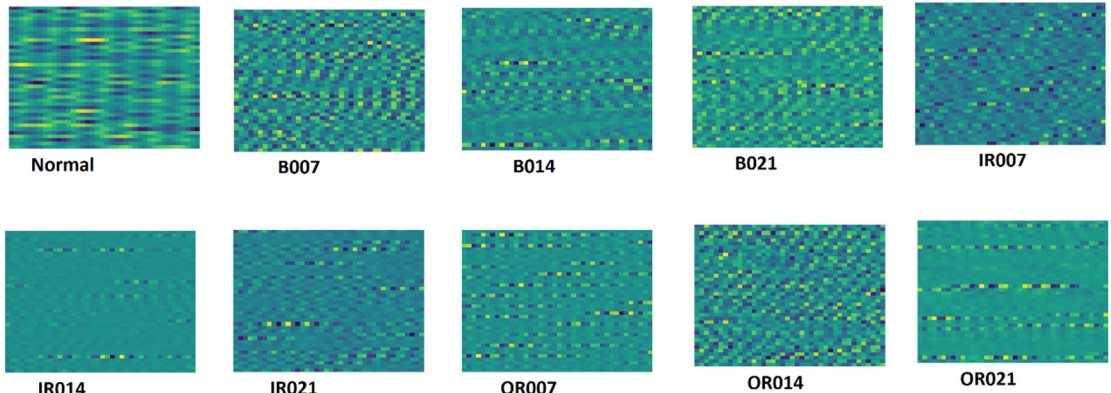

**Figure 7.** Bearing vibration images of ten different health conditions.

### 4.2. Comparison Using 2-D CNN

For the 2-D CNN, all the pre-processing steps were the same as those applied to the 1-D CNN, adding one further step. Each sample, consisting of 1600 data points, was further illustrated as a 2-D image of size $40 \times 40$ pixels. Hence, the input size for the 2-D CNN was images of shape $40 \times 40$. The network parameters implemented for the 2-D CNN were equivalent to those for the 1-D CNN. The kernel size was $3 \times 3$, equivalent to $9 \times 1$, stride size was 1, and max-pooling layer size was $2 \times 2$. The 2-D CNN architecture is shown in Figure 8 and the network parameters are listed in Table 3. The result obtained using the 2-D CNN is highlighted in Table 5.

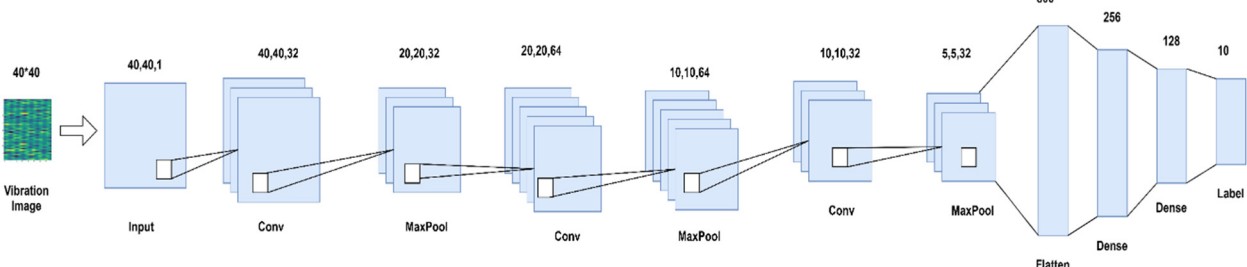

**Figure 8.** Network architecture of 2-D CNN model used for comparison.

### 4.3. Comparison of the Proposed Model with Some Other Published Works

We also compared the proposed model with recently published works using the CWRU vibration dataset. Table 6 shows the comparison.

**Table 6.** Comparison of the proposed model with some published works.

| Article Reference | Model | Accuracy |
|:---:|:---:|:---:|
| [46] | 2-D CNN using vibration image | 97.74% |
| [42] | 1-D CNN | 97.1% |
| [44] | 2-DCNN-based approach with multiple sensor fusion | 99.41% using 2 sensors 98.35% with 1 sensor |
| [47] | CNN-based deep distance metric learning method | 99.34% for sample length of 8192 |
| Our Model | 1-D CNN | 99.34% to 99.49% for 4 different datasets |

### 5. Discussion and Conclusions

In the context of the growing concepts of Industry 4.0 and smart manufacturing, intelligent methods for detecting and classifying machine faults are the subject of increasing scientific research and interest. Different signal processing and ML-based approaches have been used in detecting and classifying bearing faults. Signal processing techniques in the time domain, frequency domain, and time-frequency domain have been used to analyze vibration signals. However, due to the various limitations of the typical signal processing and ML-based approaches, DL-based methods are preferred over them. An intelligent method for bearing fault detection and classification in one of the most used benchmark datasets, the CWRU dataset, is presented in this paper. The 1D CNN-based deep learning approach is implemented for the time-sequence bearing data. The raw vibration data from four datasets are divided into *N* samples, with each sample containing 1600 data points, and then fed into a 1-D CNN for feature extraction and classification.

Moreover, a comparison of the proposed method with the 2-D CNN using 2-D image representation of the raw bearing signal as input is carried out. The result shows that the 1-D CNN performs efficiently for time-series data. In addition, sensitivity analysis of the proposed model is performed, in which metrics such as precision, recall, and f1-score are determined. We also compare the proposed method with some of the published works as well. We use Keras [50] for training the model, using TensorFlow [51] at the backend. The model is trained through the GeForce RTX 2080 Ti GPU of NVIDIA [52]. The results analysis also shows that the time for training the 1-D CNN model is lower than that for the 2-D CNN. The Results section thus shows that implementing a 1-D CNN is more efficient in terms of computational complexity for time-series data. With only 276,522 parameters, the proposed method achieves state-of-the-art accuracy. Simplicity and computational feasibility are the main advantages of this model.



**Author Contributions:** Conceptualization, D.N., J.S., Y.K. and J.H.; methodology, D.N.; software, D.N.; validation, D.N., Y.K., J.S. and J.H.; formal analysis, D.N., J.S., Y.K. and J.H.; data curation, D.N. and Y.K.; writing—original draft preparation, D.N.; writing—review and editing, D.N., Y.K., J.S. and J.H.; supervision, J.S. and J.H.; funding acquisition, J.S. and J.H. All authors have read and agreed to the published version of the manuscript.

**Funding:** This work was supported by the "Regional Innovation Strategy (RIS)" through the National Research Foundation of Korea (NRF), funded by the Ministry of Education (MOE) (2021RIS-003).

**Institutional Review Board Statement:** Not applicable.

**Informed Consent Statement:** Not applicable.

**Data Availability Statement:** The data was collected from a publicly available CWRU bearing database. The link is as follows: https://engineering.case.edu/bearingdatacenter/download-data-file (accessed on 19 October 2021).

**Acknowledgments:** The authors would like to express their sincere appreciation to Case Western Reserve University for the open-source CWRU bearing data.

**Conflicts of Interest:** The authors declare no conflict of interest.

## Abbreviations

| | |
|---|---|
| 1-D | One-Dimensional |
| 2-D | Two-Dimensional |
| ANN | Artificial Neural Network |
| CWRU | Case Western Reserve University |
| CNN | Convolutional Neural Network |
| DE | Drive End |
| DL | Deep Learning |
| FE | Fan End |
| GPU | Graphics Processing Unit |
| IR | Inner-Race |
| k-NN | k-Nearest Neighbor |
| ML | Machine Learning |
| NS | Normal State |
| OR | Outer-Race |
| PCA | Principal Component Analysis |
| RMS | Root Mean Square |
| ReLU | Rectified Linear Unit |
| SVD | Singular Value Decomposition |
| SVM | Support Vector Machine |

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
