# Peer review of "CNN-Based Fault Detection for Smart Manufacturing â€"

_applsci, doi:10.3390/app112411732_

Round 1
Reviewer 1 Report
This paper is well written, some minor problems should be addressed.
1: How about the computing time of the proposed method.
2: What is the stability of the results of the proposed method.
3: The comparisons are insufficient, please consider some state-of-the-art methods.
4: The influence of parameters is not discussed.
5: The literature review is not enough, please add some papers that are highly related to CNN, such as "Ensemble transfer CNNs driven by multi-channel signals for fault diagnosis of rotating machinery cross working conditions[J]"
Author Response
We would like to express our sincere thanks for allowing a resubmission of our paper, with an opportunity to address the reviewers' comments.
We also would like to express our sincere gratitude to the reviewers for a careful and thorough reading of this paper, and for the thoughtful comments and constructive suggestions, which, indeed, helped to improve the quality of this paper
The computing time of the proposed method is mentioned in Table 7. We also have discussed the stability of the model. We also have added the necessary texts in the literature review, compared our model with some more published works, and cited the necessary texts from the journal you have recommended.
We are open to more reviews and will improvise our manuscript again in the future if needed. Thank you for your guidance and comments.
Reviewer 2 Report
This paper presents a CNN-based fault detection scheme for bearing health condition monitoring. The benchmark CWRU data set was used to verify the proposed method. However, the experiment was unconvincing. Some recently-released results, e.g., 10.1109/TII.2020.3007653, 10.1016/j.ymssp.2020.106956, 10.1109/TIE.2019.2896109 , should be compared or at least discussed.
In addition, some references are old. Please cite more recent articles in order to increase the quality of this paper.
Author Response
We would like to express our sincere thanks for allowing a resubmission of our paper, with an opportunity to address the reviewers' comments.
We also would like to express our sincere gratitude to the reviewers for a careful and thorough reading of this paper, and for the thoughtful comments and constructive suggestions, which, indeed, helped to improve the quality of this paper.
We have added the necessary texts in the literature review, compared our model with some more published works, and cited the necessary texts from the journal you have recommended.
We are open to more reviews and will improvise our manuscript again in the future if needed. Thank you for your guidance and comments.
Reviewer 3 Report
Applied Sciences (ISSN 2076-3417)
The following is an overview of the article CNN-Based Fault Classification of Time-Sequence CWRU Bearing Data (applsci-1448232). In this study, author(s) proposed a new a deep learning-based 1D convolutional neural network is proposed using the time-sequence bearing data from the Case Western Reserve University (CWRU) bearing dataset. The manuscript has contributions to the area of Bearing Fault Diagnosis identification (Classification). The author(s) mentioned; With the rapid development of science and technology, production facilities are also growing advanced. Smart manufacturing plants are the result of intelligent systems deployed in the factory. Smart factories produce more, but this leads to increased machine failures when the machines are in daily operation and used in almost all applications. Various deep learning methods have been used and implemented to detect and diagnose bearing damage using raw data.
The author(s) stated in the first part of the study; Electrical machines are used ubiquitously in industrial applications these days. With the development and advancement in science and technology, modern industries are also developed rapidly. As a result, machinery equipment functions on a day-to-day basis and for almost every application, which makes these types of machinery work under unfavorable circumstances, excessive loads, and humidity. This results in motor failures leading to massive maintenance expenditures, deprivation in production level, severe financial losses, and possible risk of loss of lives. The rotating machinery and induction engines play a vital role in the manufacturing systems. These rotating machines consist of numerous elements, like stator, rotor, shaft, and bearings. Rolling element bearings (REBs) are generally termed as bearings and are the vital and vulnerable components in the machine whose fitness state affects the effectiveness and performance, stability and lifespan of the machinery.
The author(s) stated in the last part of the study; author(s) proposed, a deep learning-based 1D convolutional neural network is proposed using the time-sequence bearing data from the Case Western Reserve University (CWRU) bearing dataset. Four different datasets were used. The proposed method achieves state-of-art accuracy with a smaller amount of training data. Metrics such as precision, recall, and f-measure were determined for the sensitivity analysis of the proposed method. Next, we compared the proposed method with a 2D CNN that uses 2D illustration images of raw data as input. The method shows the effectiveness of using 1-D CNNs over 2-D CNNs for time-sequence data. The proposed method has a low computational cost and can beat most complex and computationally intensive algorithms previously used for bearing damage detection and diagnosis.
Finally, the authors mentioned; the results showed that the implementation of 1-D CNN is more efficient in terms of computational complexity for the time series data. Simplicity and computationally feasible are the main advantages of this model.
However, some points must be highlighted so that the author(s) can review and submit in another round of review: The following corrections are considered to be beneficial for the strengthening of the article.
- The Conclusions should be reviewed again. The original aspect of the study and its difference from other studies should be clearly explained. (The conclusion should be explored better and it needs to contemplate the eventual restrictions of the developed technique to address future works in this area.)
- The abstract must be make strong. Abstract should be reviewed again.
- Some sentences have spelling errors. (Punctuation marks, spaces, etc.). Some places should be left space. Please check all sentences in article.
- It has been a comprehensive study in the literature in recent years. If there are more current literature studies, these should be examined in detail and added to the literature section (Especially, fault diagnosis, bearing fault diagnosis and vibration signals identification studies.). It is a suggestion for the literature part of the article to be more comprehensive. It may be useful to include relevant articles in 2018-2021 in references. As an example, I think it might be useful to add the article to references, such as the articles below, to keep the article updated as a literature. (3.1) Classification of bearing vibration speeds under 1D-LBP based on eight local directional filters. Soft Computing. 3.2) A new feature extraction approach based on one dimensional gray level co-occurrence matrices for bearing fault classification. Journal of Experimental & Theoretical Artificial Intelligence 3.3) The effect of bearings faults to coefficients obtaned by using wavelet transform. 22nd Signal Processing and Communications Applications Conference (SIU) IEEE. 3.4) Application of an enhanced fast kurtogram based on empirical wavelet transform for bearing fault diagnosis. Measurement Science and Technology. 3.5) An intelligent approach for bearing fault diagnosis: Combination of 1D-LBP and GRA. IEEE Access. 3.6) Feature extraction of ball bearings in time-space and estimation of fault size with method of ANN, In Proc. 16th Mechatronika)
- The authors should compare the results of their method with those of previous studies. As mentioned in the literature, there are several methods with very high accuracy, even better than the proposed method. Author(s) can do compare table (A new table can add about previous studies to result section.). They can investigate this article (https://doi.org/10.1016/j.asoc.2019.106019) and they can add comparison table about Fault Classification. Authors can use that article (It must give reference). This subject is very important.
- The motivations of the proposed method are not clear. Which problem does the proposed method attempt to solve? Why the other existing diagnosis methods failed to solve it? What are the advantages of the proposed method compared to other methods? Those should be illustrated more clearly.
- Carefully check all grammatical error. Still, the English language should be improved. I suggest asking for help from a native English.
- If it is possible to increase the image quality of Figure 6, it should be rearranged.
Author Response
We would like to express our sincere thanks for allowing a resubmission of our paper, with an opportunity to address the reviewers' comments.
We also would like to express our sincere gratitude to the reviewers for a careful and thorough reading of this paper, and for the thoughtful comments and constructive suggestions, which, indeed, helped to improve the quality of this paper.
We have edited the abstract, the conclusion, added the motivation. We also have checked the grammatical mistakes and checked the English language.
Since figure 6 (now figure 7) is the combination of each picture of size 40*40, the image is of low quality. We are afraid to tell you that we could not do anything more to improvise its quality. However, all the other figures are vectorized. We also have uploaded the zip files of the figures. We have included the recent research as you recommended and removed the old ones. However, we have included some highly remarkable and benchmark studies though they are old.
We have added the necessary texts in the literature review, compared our model with some more published works, and cited the necessary texts from the journal you have recommended. We also have uploaded the highlighted manuscript showing the changes we have made.
We are open to more reviews and will improvise our manuscript again in the future if needed. Thank you for your guidance and comments.
Round 2
Reviewer 2 Report
This paper can be accepted.